# Effect of Prepartum Maternal Supplementation with Diphenyl Diselenide on Biochemical, Immunological, and Oxidative Parameters of the Offspring

**DOI:** 10.3390/ani14010010

**Published:** 2023-12-19

**Authors:** Cláudia Medeiros Rodrigues, Eliana Burtet Parmeggiani, Karoline Wagner Leal, Marla Schneider, Silvana Giacomini Collet, Francielli Weber Santos Cibin, Viviani Gomes, Maiara Garcia Blagitz, João Batista Teixeira da Rocha, Marta Lizandra do Rêgo Leal

**Affiliations:** 1Department of Large Animal Clinic, Federal University of Santa Maria, Santa Maria 97105-900, RS, Brazil; elianabparmeggiani@hotmail.com (E.B.P.); karolwagnerleal@gmail.com (K.W.L.); 2Veterinary Medicine Course, Central Education Unit Faem Faculty, Chapecó 89812-214, SC, Brazil; marlaschneideruffs@gmail.com; 3Veterinary Medicine Course, University of West Santa Catarina, Xanxerê 89820-000, SC, Brazil; silvana.collet@unoesc.edu.br; 4Oxidative Stress Laboratory, Federal University of Pampa, Uruguaiana 97501-970, RS, Brazil; franciellicibin@unipampa.edu.br; 5Department of Clinical Medicine, Faculty of Veterinary Medicine and Zootechnics, University of São Paulo, São Paulo 05508-270, SP, Brazil; viviani.gomes@usp.br; 6Veterinary Medicine Course, Federal University of the Southern Border, Realeza 85770-000, PR, Brazil; maiara.azevedo@uffs.edu.br; 7Department of Biochemical and Molecular Biology, Federal University of Santa Maria, Santa Maria 97105-900, RS, Brazil; jbtrocha@yahoo.com.br

**Keywords:** cattle, diphenyl diselenide, immunity, supplementation

## Abstract

**Simple Summary:**

The rearing of calves is a challenging phase in the production of dairy cattle. These animals face challenges from birth, as they require ingestion of colostral immunoglobulins for transfer of passive immunity, given that their immune systems are still immature. If there is a failure in the transfer of passive immunity, they become vulnerable to disease. This vulnerability, combined with stressors such as weaning, can predispose them to cellular oxidative stress. Similarly, to the immune system, the antioxidant defense system is also in the developmental stage, therefore, providing antioxidant substances during this phase could be a valuable option. Considering this, our team proposed to assess the impact of prepartum maternal selenium supplementation on the offspring, given its established benefits and its transfer through the placenta, colostrum, and milk. Diphenyl diselenide, an organic analog, was selected because of its anti-inflammatory and antioxidant properties. It can be administered safely subcutaneously, ensuring the correct dosage and facilitating its use at strategic times without any risk of toxicity. Maternal supplementation with diphenyl diselenide has been found to significantly increase passive immunity transfer rates in calves, as well as help establish and improve humoral immunity. This provides new insight into the immunomodulatory capabilities of this molecule.

**Abstract:**

This study aimed to assess the impact of prepartum maternal diphenyl diselenide (PhSe)_2_ supplementation on the development, biochemical, immune, and antioxidant parameters of calves. Eighteen Holstein breed calves were used, born to females who were or were not subjected to supplementation, at 42, 28, and 14 days prior to calving. The (PhSe)_2_ group (DDG) was administered 3 μmol/kg of (PhSe)_2_ in 4 mL of dimethyl sulfoxide (DMSO), while the DMSO and NaCl groups were administered 4 mL of DMSO and 0.9% NaCl, subcutaneously. The calves were evaluated based on their weight, withers height, body condition score 24 h post-birth (0), as well on days 14, 28, 42, 56, 70. Blood samples were also taken to determine serum variables. Calves on the DDG showed higher average levels of total protein, albumin, and globulins on day 0, and the immunoglobulin G level was significantly higher than the other groups on days 0, 14, 56, 70. Maternal supplementation showed immunomodulatory effect on calves, evidenced by the exceptional rates of passive immunity transfer, as well as the enhancement of humoral immunity. Our research offers fresh insights into the immunomodulatory potential of (PhSe)_2_, making it a viable alternative in facing this challenging phase, rearing dairy calves.

## 1. Introduction

The period extending from the neonatal phase to the weaning of calves is considered a critical stage in dairy cattle production, given the numerous challenges encountered during this timeframe. First, these animals depend on the proper passive transfer of maternal colostral immunoglobulins for their protection since their immune system continues to develop even after birth. If the colostration process is not executed correctly, there will be a failure in the transfer of passive immunity [1]. This condition increases the vulnerability of animals to diseases that typically arise during this period, such as diarrhea, pneumonia, and omphalopathy [2]. These diseases are strongly associated with calf mortality as well as weight loss and/or reduced weight gain in the surviving animals [3].

The intrauterine conditions to which the fetus is exposed during pregnancy significantly impact the immune function, health, and even milk production [4]. These factors have been related to the occurrence of metabolic and oxidative stress (OS), triggered by mother’s exposure to heat stress, and restricted or excessive energy intake [5]; in humans there are already reports of cases of the restriction of intrauterine growth triggered by OS [6]. Furthermore, shortly after birth, calves are exposed for the first time to an environment rich in O_2_. As soon as the breathing process begins, there is an immediate increase in the production of reactive oxygen species [7].

The weaning process represents another critical juncture, as it can induce metabolic stress, causing the animal to deplete its energy and protein reserves, culminating in the onset of OS. Oxidative stress is also related to the incidence of diseases [4] and various stressors that animals undergo from the neonatal stage to weaning, which include everything from prophylactic health measures, castration, dehorning, etc. [8]. Given that their immune and antioxidant defense systems are still immature [9], these animals are more susceptible to the occurrence of diseases and the establishment of oxidative stress.

The use of mineral supplementation and its impact on animal health and development have been a major focus of study in veterinary medicine in recent years. Among these elements, we particularly emphasize the importance of selenium (Se). The role of this trace mineral in maternal health and its impact on offspring deserves significant attention. When Se is provided to mothers during pregnancy, it can adequately meet the needs of the fetus and the newborn. This is because it is efficiently transferred to the newborn through the placenta, colostrum, and milk [10,11,12,13]. Furthermore, Se status has been established to play a crucial role in improving the rate of passive immunity transfer provided to calves through the consumption of maternal immunoglobulins found in the colostrum [14].

However, in most of the studies mentioned above, Se was provided to the animals through their diet. It is known that, when Se is administered orally to ruminants, absorption occurs primarily in the duodenum, where it can be reduced to insoluble compounds. Therefore, Se absorption in ruminants by this method is considered less effective compared to non-ruminant’s absorption capacity [15]. Therefore, parenteral administration of diphenyl diselenide (PhSe)_2_ provides a viable alternative to the use of more traditional forms of Se available on the market [16]. Our research team investigated the distribution of this compound in sheep. The results showed that its subcutaneous application is safe, even when administered in varying doses. This is because diphenyl diselenide did not induce any signs of toxicity in animals subjected to various treatments [17]. Based on the information provided in this study, we began an investigation of the use of this molecule as an alternative mineral supplement to that typically used. The aim is to use it during challenging and strategic periods within various ruminant production systems.

Diphenyl diselenide (PhSe)_2_ is an organic Se analog with numerous pharmacological properties previously documented in the literature. Its therapeutic potential is derived from its antioxidant characteristics [18], and its importance for human health has also been confirmed due to its antioxidant and anti-inflammatory properties [19]. However, in studies involving various animal species, (PhSe)_2_ has shown numerous functions, including cardioprotective properties by reducing hypercholesterolemia and OS in rabbits [20] and inhibiting the oxidation of low-density lipoproteins in vitro [21]. It has also shown hepatoprotective properties [22] and thwarted OS triggered by septicemia in rats [23].

Furthermore, it showed a neuroprotective effect against methylmercury-induced toxicity in mice [24] and showed fungicidal properties in rabbits infected with *Pythium insidiosum* [25]. (PhSe)_2_ antioxidant properties have been clarified in studies of fish exposed to herbicides [26] and in studies of quails, as evidenced by the superior quality of their meat [27]. It has also shown immunomodulatory effects and anti-inflammatory potential through experimental models of parasitic infections and liver damage in mice [28,29].

Despite numerous studies showing the beneficial effects of diphenyl diselenide (PhSe)_2_ on various animal species, its use in livestock is rarely explored in research. In dairy cattle, studies are yet to evaluate whether supplementing pregnant females with (PhSe)_2_ significantly affects the health and development of their calves. Therefore, this study aimed at assessing the impact of maternal supplementation with (PhSe)_2_, strategically administered during the prepartum period; the development, biochemical, and immune parameters; and the oxidative status of Holstein calves from the neonatal stage to weaning.

## 2. Materials and Methods

### 2.1. Ethic Committee on Animal Use

The animal study protocol was approved by the Ethic Committee on Animal Use of the Federal University of Santa Maria (UFSM) under the process number 8663120721 on 5 October 2021.

### 2.2. Animals, Experimental Design, and Diet

The study involved eighteen Holstein calves (*n* = 18, consisting of eleven females and seven males), with an age range of 1–70 days old, specifically from the neonatal period to the weaning stage. These animals are the offspring of eighteen cows that received or did not receive diphenyl diselenide (PhSe)_2_ supplementation during the prepartum period. The eighteen dams chosen for the study were multiparous Holstein cows, with a body condition score ranging from 3 to 3.75 (where 1 indicates thin and 5 indicates obese) and an average daily milk production of approximately 35 L during their previous lactation period.

Initially, the cows underwent a physical examination [30] prior to inclusion in the study to evaluate their health status. This ensured that animals with clinical disorders were not selected. No changes were implemented in the management or diet of the animals; therefore, they adhered to the standard farm routine. They were housed in a compost barn system throughout the prepartum period, and their diet consisted of corn silage, ryegrass hay, concentrated feed, vitamin and mineral supplements (Table 1), and water provided ad libitum. Furthermore, there were no complications during pregnancy, and all calves were born during normal delivery.

The eighteen (*n* = 18) calves involved in the study were part of the three groups that correspond to the treatments their mothers received. These eighteen females were subjected to three different treatments on days 42, 28, and 14 before giving birth. Six cows in the diphenyl diselenide (DDG) group were administered 3 μmol/kg of (PhSe)_2_ diluted in 4 mL of dimethyl sulfoxide (DMSO) (*n* = 6), while the DMSO (*n* = 6) and NaCl (*n* = 6) groups were administered 4 mL of DMSO and a 0.9% NaCl solution, respectively. All treatments were administered via subcutaneous (SC) route. Therefore, six calves (three females and three males) born to mothers who received (PhSe)_2_ supplementation were assigned to the DD group (*n* = 6), while the rest belonged to the DMSO group (DMSOG) (*n* = 6), consisting of three females and three males, and the NACL group (NACLG) (*n* = 6), which included five females and one male.

The calves were accommodated in individual suspended stalls immediately after birth and remained there for the entire duration of the experimental period. Colostrum was administered with fresh colostrum, obtained from their respective mothers, in a volume equivalent to 15% of their live weight within the first 6 h of life. The colostrum was assessed using a Brix refractometer (model BTX-1, Vee Gee^®^ Scientific, Vernon Hills, IL, USA), and immunoglobulin G (IgG) concentrations (Table 2) were determined using a sandwich ELISA [31]. 

The diet consisted of providing a 6-L bottle of milk at 37 °C, divided into two servings per day, together with a concentrate, vitamin and mineral supplements (Table 3), and water ad libitum. Furthermore, ryegrass hay (*Lolium multiflorum*) was available to animals from 10 days of age. The calves, along with their mothers, did not undergo any changes in diet or management. The pre-established routine on the farms was followed throughout the study.

### 2.3. Diphenyl Diselenide (PhSe)_2_


(PhSe)_2_, 98% were obtained from Sigma-Aldrich^®^ (catalog no. 180629, St. Louis, MO, USA) and diluted in dimethyl sulfoxide (DMSO) (C_2_H_6_SO; 99.2%) at the time of administration.

### 2.4. Execution Period and Location

The study was carried out on two commercial dairy farms from October 2021 to August 2022, with a specified execution period between December and March, in response to the observed heat stress during the summer months. The farms are located in the western region of Santa Catarina state, Brazil, known for its humid subtropical climate.

### 2.5. Evaluation and Sample Collection 

The calves underwent weighing, measurement of withers height, and evaluation of the body condition score (BCS) 24 h after birth (day 0) on days 14, 28, 42, 56, and 70 of their life. Blood samples were collected at the same intervals via jugular venipuncture with a 21 G needle using a vacuum collection system (BD Vacutainer^®^, Franklin Lakes, NJ, USA) in a tube without an anticoagulant (9 mL).

### 2.6. Blood Tests

Serum biochemical variables, such as total protein (TP), albumin (ALB), and gamma-glutamyl transferase (GGT), were identified using an automatic biochemical analyzer (BS 230, Mindray^®^, Mahwah, NJ, USA) and commercial kits (Bioclin^®^, Belo Horizonte, MG, Brazil). The values of globulin (GLOB) and ALB:GLOB ratio (A:G) were determined by calculating the difference between the TP and ALB values (TP-ALB = GLOB) and by dividing the ALB by GLOB fractions (ALB/GLOB = A:G), respectively.

Immunoglobulin G concentrations were assessed in serum samples with sandwich ELISA [31]. The levels of thiobarbituric acid reactive substances (TBARS) [32] and reduced glutathione (GSH) [33] were also measured in serum samples.

### 2.7. Statistical Analysis

The assumptions of normality, homoscedasticity, and independence of residuals were previously tested using the Shapiro–Wilk test. Variables that did not have a normal distribution (GGT and IgG) were transformed. The Box-Cox transformation was applied to the GGT variable (λ = −0.5), and the natural logarithm transformation was performed on the IgG variable [34]. The response variables were analyzed using a linear model, with repeated measurements over time, satisfying the following statistical model: Y_ijkl_ = µ + TREAT_i_ + MOMENT_j_ + (TREAT × MOMENT)_ij_ + ε_ijkl_; where: Y_ijkl_ = response variables; μ = general average of all observations; TREAT_i_ = treatments (NACLG, DMSOG, DDG); MOMENT_j_ = evaluation moments (0, 14, 28, 42, 56, and 70 days of life); (TREAT × MOMENT)_ij_ = effect of the interaction between treatments and the moment of assessment; ε_ijkl_ = random error associated with each observation, being ~NID (0, $\sigma_{\varepsilon }^{2}$). If there was a statistical difference, the Tukey mean comparison test was performed. For this purpose, the statistical program R Studio^®^ (R CORE TEAM, 2013, Vienna, Austria) was used with a significance level of 5%. Data were presented as averages and standard errors.

## 3. Results and Discussion

To find viable solutions to alleviate the challenges faced by calves from the neonatal stage to weaning, this study implemented maternal supplementation with organic Se, specifically in the form of (PhSe)_2_, during the prepartum period. The study then evaluated the impact of this supplementation on the health and development of calves. It is widely recognized that calves face numerous challenges from the moment of birth, including adapting to the environment outside the womb and the need to acquire immunoglobulins by consuming maternal colostrum. These factors, coupled with a still developing immune system, make them susceptible to diseases that typically manifest during the breastfeeding stage [35].

Se is a vital component of 20 to 25 selenoproteins in vertebrates [36], in particular, iodothyronine deiodinase, thioredoxin reductase (TrxR), and glutathione peroxidase (GPx) [37]. The TrxR and GPx enzymes perform antioxidant functions, while iodothyronine deiodinase controls the production of the active T3 hormone (triiodothyronine) in peripheral tissues and the thyroid, which contributes to animal metabolism and growth [38]. Therefore, Se plays a crucial role in the development of young animals, particularly during the initial stages of growth [39]. This fact underscores the importance of this element for calves.

The monitored development parameters showed that the animals in the NaCl group experienced a greater weight gain (Figure 1; *p* = 0.002) compared to those in the DMSO group on days 42, 56, and 70 of their life. There were no statistical differences between the groups regarding withers height (Figure 1; *p* > 0.05), but the BCS (Figure 1) showed interaction between treatments (*p* = 0.020), with the NACLG presenting lower values for this evaluation in two moments (days 0 and 14) when compared with DMSOG. However, a significant increase in body weight (*p* < 0.001) and height at the withers (*p* < 0.001) was observed on different days within all groups for almost the entire experimental period. In the BCS assessment, there was a significant increase from days 0 and 14 to 70 within the NaCl group (*p* < 0.001). We believe that these results stem from the uniform development exhibited by the calves involved in the study, which occurred regardless of the treatment their mothers underwent. In another study, our research team observed positive outcomes in the weaning weight (on the 70th day of life) of female Holstein calves when (PhSe)_2_ was administered from birth until weaning [40], similarly to a previous study [41], which also reported a significant weight gain on the 270th day of life after administering (PhSe)_2_ to Holstein calves aged 50 to 70 days. However, in both studies, the administrations were performed directly on the calves, unlike in this study, where we used maternal supplementation with (PhSe)_2_ during the prepartum period. On the other hand, when we compare the immunological responses triggered by maternal supplementation and direct administration to calves, the results are similar. Because we found a significant increase in IgG in calves born to cows subjected to supplementation with (PhSe)_2_ and because of results from a previous study, calves subjected to supplementation also showed an increase in this variable [40].

In this study, total protein, albumin, gamma-glutamyl transferase, and immunoglobulin G measurements were performed, as these variables are critical to evaluate the success or failure of passive immunity transfer in calves. These animals are known to have agammaglobulinemia due to the characteristics of the syndesmochorial placenta of female bovines. This placenta separates maternal and fetal blood supplies, preventing the transfer of immunoglobulins. As a result, newborns rely on maternal colostral immunoglobulin ingestion for their protection immediately after birth [2].

The DD group showed interaction treatment x moment to TP (Figure 2; *p* = 0.049) and GLOB (Figure 2; *p* = 0.038). Furthermore, DDG showed the highest average levels of TP (*p* = 0.002), ALB (Figure 2; *p* = 0.043), and GLOB (*p* = 0.007) on day 0 compared to the other experimental groups. The same pattern can be observed in the TP (*p* = 0.003) and GLOB (*p* < 0.001) values within the DD group when comparing day 0 with other experimental time points (14, 28, 42, 56, and 70). In the evaluation of ALB within the groups, the lowest averages were observed near the birth date (day 0). The DD group showed a significant increase in ALB (*p* < 0.001) from day 0 to days 42, 56, and 70 and again from day 14 to days 56 and 70. The A:G ratio (Figure 2) did not indicate any differences between treatments (*p* > 0.05). However, there was a significant increase (*p* < 0.001) within the groups from birth (day 0) to subsequent time points (14, 28, 42, 56, and 70 days) and in the DD group from day 14 to days 42, 56, and 70.

The significant findings of TP, ALB, and GLOB in the DD group, observed immediately after colostrum feeding, were attributed to the absorption of colostral immunoglobulins, which increased subsequently in protein fractions. When we collectively examined the TP results and serum IgG concentrations on day 0, it can be said that all calves involved in the study achieved excellent levels of passive immunity transfer. Because the proposed standard for evaluation of immunity passive transfer includes four categories: excellent (IgG ≥ 25.0 g/L and TP ≥ 6.2 g/dL), good (IgG 18.0–24.9 g/L and TP 5.8–6.1 g/dL), fair (IgG 10–17.9 g/L and TP 5.1–5.7 g/dL), and poor (IgG < 10.0 g/L and TP < 5.1 g/dL), these categories can be applied to individual calves and to the operation for herd-based evaluation based on the percentage of calves that should be represented in each category [2]. The quality of the provided colostrum also contributed to the positive outcomes observed since all mother groups showed colostral brix and IgG values considered high quality (Table 2), specifically a brix index ≥ 22% and an IgG concentration ≥ 50 g/L [2]. 

A GGT measurement has been used as an alternative and indirect method to assess the transfer of passive immunity. This is because this enzyme is found in high concentrations in the colostrum of ruminants; therefore, its activity increases in calves that have received the appropriate intake of colostrum [42]. In our study, GGT concentrations (Figure 2) did not show significant differences between the groups (*p* > 0.05). However, when comparing days, the NACL and DD groups showed the highest concentrations on day 0 (*p* < 0.001) compared to days 28, 42, 56, and 70. These results were expected due to the colostrum feeding protocol that the animals underwent during the experimental period. When we collectively evaluate the TP, GGT, and IgG data from the calves, we can affirm that passive immunity transfer was carried out successfully.

IgG levels (Figure 2) in the DD group were significantly higher (*p* < 0.001) than those in the other groups (NACL and DMSO) on days 0, 14, 56, and 70. In the DD group, there was a significant decrease in their concentrations from day 0 to days 28 and 42, followed by a significant increase from days 28 and 42 to day 70 (*p* = 0.008). We believe that (PhSe)_2_ contributed to the increase in Se bioavailability in the bodies of cows and, consequently, in their calves through the transplacental pathway and through the ingestion of colostrum with higher concentrations of this mineral due to the supplementation received by the mothers. 

Despite already being described in the literature that providing organic and, thus, more bioavailable forms of selenium to pregnant females improves the Se status of calves [43,44,45], until now, (PhSe)_2_ was used in few studies as an organic source for selenium supplementation in cattle. Therefore, it is necessary to conduct more research to evaluate the bioavailability, nutritional value, deposition, and body functions of alternative Se sources in different breeds and physiological stages of cattle raised in areas with different Se levels [46]. Selenium bioavailability also improves the absorption efficiency of IgG, subsequently increasing its serum concentration [47]. This is because Se availability is closely related to the process of pinocytosis, the mechanism by which immunoglobulins are absorbed into the intestinal epithelium [48].

It is also interesting to note that Se (as component of specific selenoproteins) has a direct role in the process of endocytosis in macrophages [49]. Accordingly, recent studies with different types of formulations containing Se have indicated a positive effect of supplementation in both pinocytosis and phagocytosis in macrophages and other mammalian cell types [50]. Of particular health importance to the calves and to our present findings, the supplementation of Se (in alfalfa Se-enriched diets) to the dams improved the pinocytosis of newborn calves (as determined by the passive transfer of ovoalbumin) in a concentration-dependent manner [14]. In accordance, the addition of selenium to the colostrum given to calves just after the partum significantly increased the calves IgG level for up two weeks; the authors suggested that Se supplementation can improve the pinocytosis in new born calves, and this specific case shows that the effect of Se on IgG absorption is not only nutritional but also has pharmacological potential [47]. When (PhSe)_2_ was administered to Holstein calves, we observed a significant increase in IgG concentrations from the neonatal stage to weaning. Furthermore, we observed a reversal in the IgG trend. In the control group, there was a gradual decrease in age progression, while the treated animals showed an increase [40].

When examining IgG values among all groups, we observed a similar pattern between them, marked by a significant drop at 28 and 42 days of life. This physiological decline, known as the immunological window, occurs due to a decrease in the concentration of immunoglobulins ingested through the colostrum, which are responsible for providing passive immunity to the newborn. This decrease is likely due to the plasma half-life of IgG, approximately 21 days [51]. Simultaneously, the active immune system of the calf has not yet reached full maturity [52].

Shortly after this decline, the DD group exhibited a substantial increase in IgG concentrations compared to the other groups at 56 and 70 days of age. We attribute this increase to the previously documented ability of (PhSe)_2_ to accumulate in deeper tissues, such as the liver, kidneys, and brain, and gradually distribute in the bloodstream [17]. Therefore, we believe that (PhSe)_2_ remained in storage sites and, through its continuous release, contributed to the development of humoral immunity in calves. This is because IgG constitutes 85% of the γ globulin fraction, which is composed of immunologically active substances (immunoglobulins A, M, E, and G) and plays a crucial role in antibody formation [53].

Immunoglobulins, produced by plasma cells, are considered the primary components of humoral immunity. Furthermore, research suggests that Se can enhance the synthesis of antibodies and immunoglobulins, indicating its significant role in IgG production [46]. Additionally, the stimulation of immune function initiated by this mineral through the proliferation of T cells and activation of natural killer (NK) cells is also well established [54]. This leads us to conclude that organic Se supplementation, i.e., (PhSe)_2_, improved the immune function of the animals. Another crucial aspect is that the use of organic Se sources is more effective than that of inorganic sources. This is because organic Se has a greater capacity to enhance the expression of hepatic and muscular selenoprotein genes [55]. In addition, it can efficiently stimulate lymphocytes to secrete cytokines, signaling the initiation of humoral immunity and, subsequently, leading to the production of immunoglobulins [56].

Although the exact mechanisms involved in the increase in IgG levels in calves caused by (PhSe)_2_ administration to the dams is unknown, the immunostimulatory effects of different chemical forms of selenium have been frequently described in the literature (for a comprehensive review see [49] and the references therein). In effect, the stimulation of different selenoprotein synthesis by diphenyl diselenide [57] can contribute to the immunostimulant effects observed in Holstein calves born from supplemented cows. In addition, by modulating the anti-inflammatory and antioxidant pathways [58], diphenyl diselenide can also indirectly improve the general health of calves.

Elevated levels of TBARS (Figure 3) were observed in the DD group on day 0 compared to the other groups (*p* = 0.010). The DD group showed the highest concentrations on the day of birth (day 0), followed by a significant reduction (*p* < 0.001) on days 28, 42, 56, and 70. The elevated lipid peroxidation activity observed in the DD group on day 0 is probably the result of stress induced by birth, resulting from the initial exposure to the extrauterine environment. However, it should be noted that while minerals can exhibit antioxidant properties, they can also function as pro-oxidants [9]. We understand that the Se requirement, as stated in the literature for all categories of dairy cattle, is 300 µg/kg of dry matter (DM) [59]. However, in this study, we are unaware of the Se status of the cows at the time of administration. We must also consider the possibility that supplementation with (PhSe)_2_ may have helped generate reactive species in the DD group on day 0.

One of the primary therapeutic characteristics of (PhSe)_2_ is derived from its antioxidant potential, as it simulates the activity of the GPx enzyme [18]. In this study, we measured the concentrations of GSH since it is crucial for GPx to decompose hydrogen peroxide into water. GSH serves as an electron donor, producing oxidized glutathione (GSSG). To maintain its antioxidant activity, GSSG is reconverted to GSH through the action of glutathione reductase (GR) [60]. Therefore, by quantifying GSH, we indirectly infer the activity of GPx. However, GSH (Figure 3) did not show significant differences between treatments and between different days within the groups (*p* > 0.05). However, we did not identify a specific antioxidant response to maternal prepartum supplementation with (PhSe)_2_. One point that we have to highlight here is the absence of reference values for TBARS and GSH for this specific animal category. The absence of clinically validated normal values for those markers limit the importance of our data related to them. Furthermore, it would be important to develop the standard values of those two biochemical biomarkers of the redox state of Holstein calves.

## 4. Conclusions

Maternal supplementation during the prepartum period with organic Se, administered in this study as diphenyl diselenide (PhSe)_2_, contributed to the outstanding rates of passive immunity transfer observed and to the initiation and improvement of calves’ humoral immunity. Our findings are based on the detected concentrations of TP, ALB, GLOB, and IgG in the DD group immediately after colostrum collection and on the behavior of IgG, which showed elevated levels compared to other groups from the neonatal stage to weaning, with a physiological decrease observed only during the immunological window period.

Therefore, our research offers fresh insights into one of the numerous effects of (PhSe)_2_, specifically its immunomodulatory potential. This characteristic increasingly positions it as a viable alternative to address this challenging phase of the dairy cattle production process, specifically calf rearing. We highlighted that, in our study, (PhSe)_2_ was administered subcutaneously, ensuring that the animals received the prescribed dosage accurately. We also emphasize its safe use because it can be easily used without any risk of toxicity during the challenging and strategic phases of the production system. Furthermore, our findings also showed the importance of providing this trace mineral to pregnant cows, particularly during the prepartum phase, and its capacity to be effectively absorbed by calves through the placenta and colostrum, thus significantly improving their immune system.

## Figures and Tables

**Figure 1 animals-14-00010-f001:**
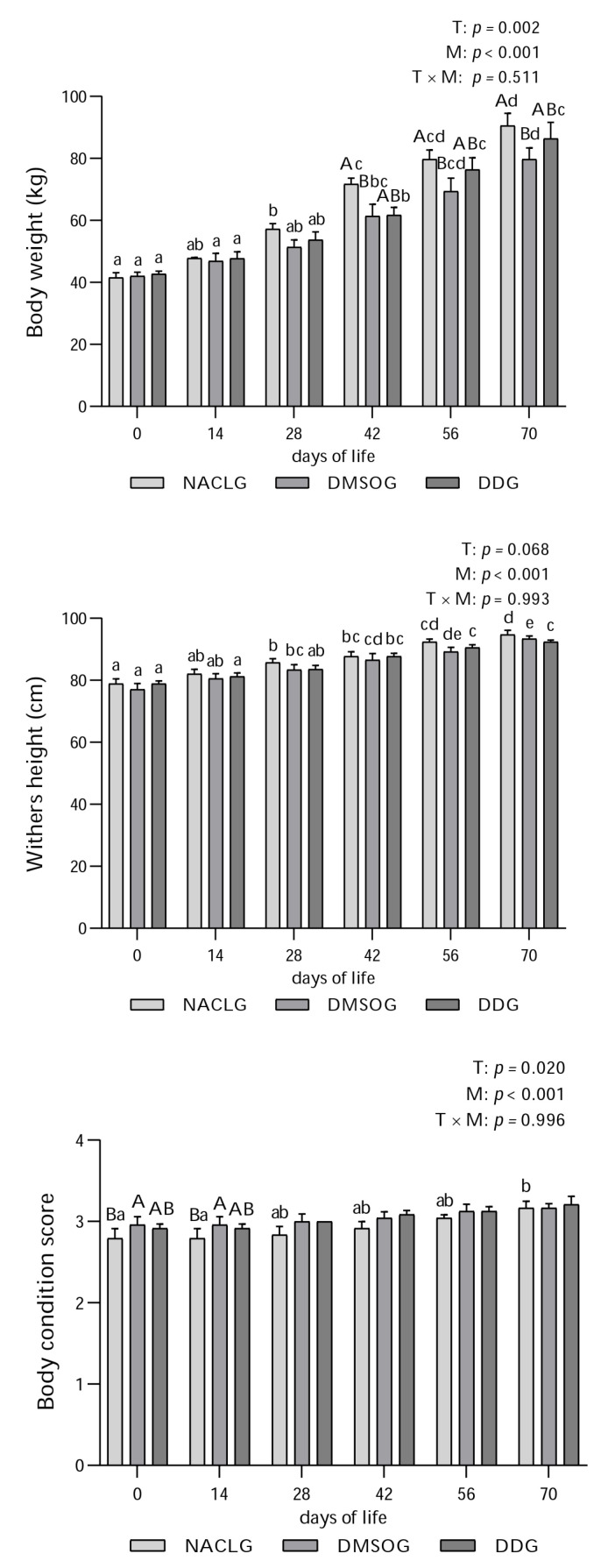
Average ± standard error and *p* value of body weight (kg), withers height (cm), and body condition score of Holstein calves from birth (day 0) to 14, 28, 42, 56, and 70 days of life, born to females who were or were not submitted to supplementation with diphenyl diselenide at 42, 28, and 14 days prior to calving. The calves were distributed into groups: diphenyl diselenide (DDG), dimethyl sulfoxide (DMSOG), and NaCl (NACLG) (average ± standard error, *n* = 6/treatment). Distinct capital letters refer to differences (*p* < 0.05) between groups on the respective days, while lowercase letters refer to differences (*p* < 0.05) between days within groups. T = treatment effect, M = moment effect, and T × M = interaction treatment moment.

**Figure 2 animals-14-00010-f002:**
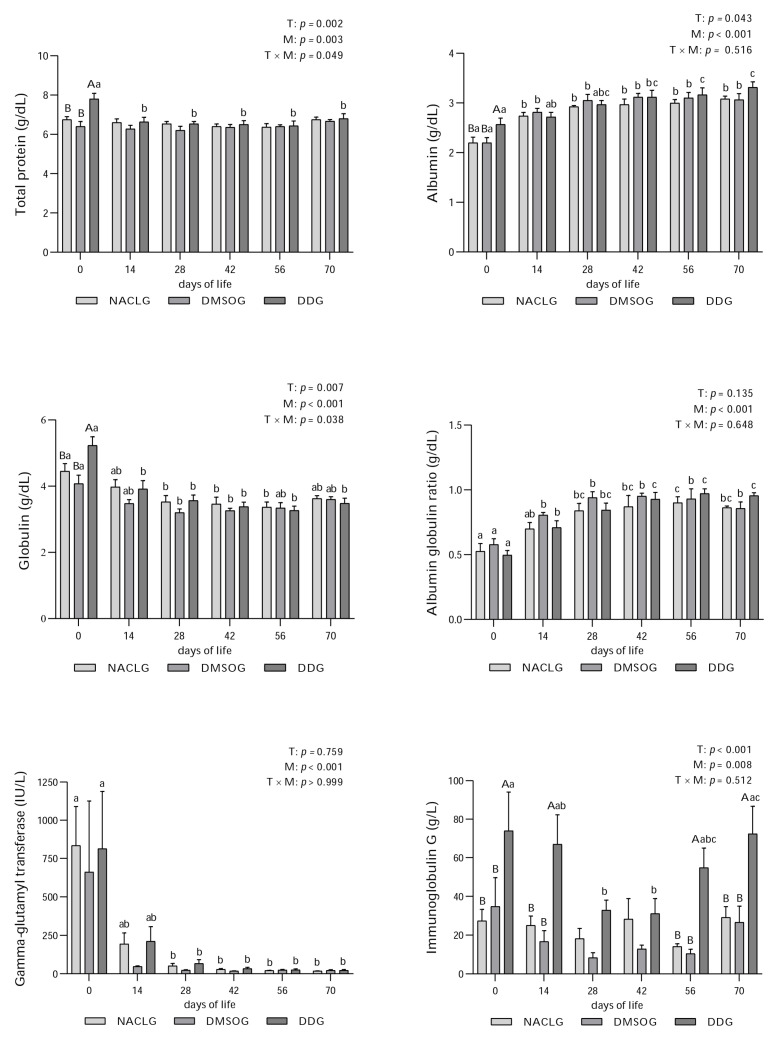
Average ± standard error and *p* value of total protein (g/dL), albumin (g/dL), globulin (g/dL), albumin globulin ratio, gamma-glutamyl transferase (IU/L), and immunoglobulin G (g/L) of Holstein calves from birth (day 0) to 14, 28, 42, 56, and 70 days of life, born to females who were or were not submitted to supplementation with diphenyl diselenide at 42, 28, and 14 days prior to calving. The calves were distributed into groups: diphenyl diselenide (DDG), dimethyl sulfoxide (DMSOG) and NaCl (NACLG) (average ± standard error, *n* = 6/treatment). Distinct capital letters refer to differences (*p* < 0.05) between groups on the respective days, while lowercase letters refer to differences (*p* < 0.05) between days within groups. T = treatment effect, M = moment effect, and T × M = interaction treatment moment.

**Figure 3 animals-14-00010-f003:**
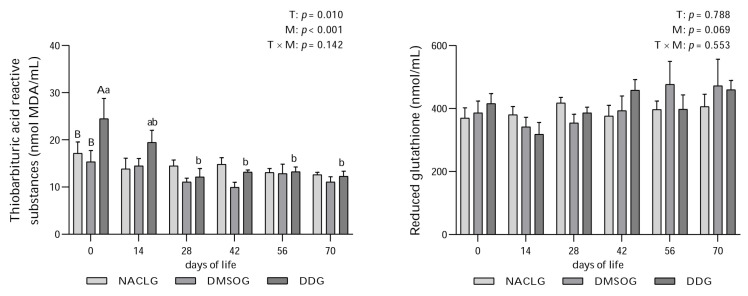
Average ± standard error and *p* value of thiobarbituric acid reactive substances (nmol MDA/mL) and reduced glutathione (nmol/mL) of Holstein calves from birth (day 0) to 14, 28, 42, 56, and 70 days of life, born to females who were or were not submitted to supplementation with diphenyl diselenide at 42, 28, and 14 days prior to calving. The calves were distributed into groups: diphenyl diselenide (DDG), dimethyl sulfoxide (DMSOG), and NaCl (NACLG) (average ± standard error, *n* = 6/treatment). Distinct capital letters refer to differences (*p* < 0.05) between groups on the respective days, while lowercase letters refer to differences (*p* < 0.05) between days within groups. T = treatment effect, M = moment effect, and T × M = interaction treatment moment.

**Table 1 animals-14-00010-t001:** Composition of the diet fed to pre-partum Holstein cows who were or were not submitted to supplementation with diphenyl diselenide at 42, 28, and 14 days prior to calving and distributed in groups: diphenyl diselenide (DDG), dimethyl sulfoxide (DMSOG), and NaCl (NACLG).

Ingredient	Values
Corn silage (% DM ^2^)	64.07
Ryegrass hay (% DM ^2^)	12.00
Soybean meal (% DM ^2^)	11.23
Soybean hulls (% DM ^2^)	10.63
Premix of vitamins and minerals ^1^ (% DM ^2^)	2.07
Chemical composition	
DM ^2^ (%)	49.8
CP ^3^ (% DM ^2^)	11.3
ADF ^4^ (% DM ^2^)	26.5
NDF ^5^ (% DM ^2^)	42.6
Starch (% DM ^2^)	23.3
Fatty acids (% DM ^2^)	3.42
Ca ^6^ (% DM ^2^)	0.51
P ^7^ (% DM ^2^)	0.31
ME ^8^ (Mcal/kg)	2.57
NEL ^9^ (Mcal/kg)	1.70
Se ^10^ (mg/kg)	0.41

^1^ Minimal vitamin and mineral levels per kg of product: vitamin A (480,000 UI); vitamin D3 (200,000 UI); vitamin E (12,000 UI); calcium (106 g); phosphorus (30 g); sulfur (90 g); magnesium (20 g); sodium (31 g); chlorine (130 g); cobalt (12 mg); cooper (600 mg); chromium (30 mg); iron (600 mg); iodine (60 mg); manganese (1600 mg); selenium (16 mg); zinc (2400); biotin (80 mg); *Saccharomyces cerevisiae* (1.5 × 109); monensin (500 mg); fluorine (300 mg). ^2^ Dry matter (DM); ^3^ Crude protein (CP); ^4^ Acid detergent fiber (ADF); ^5^ Neutral detergent fiber (NDF); ^6^ Calcium (Ca); ^7^ Phosphorus (P); ^8^ Metabolizable energy (ME); ^9^ Net energy for lactation (NEL); ^10^ Selenium (Se).

**Table 2 animals-14-00010-t002:** Average values ± standard error of colostrum brix index (%) and immunoglobulin G (IgG; g/L) of Holstein cows who were or were not submitted to supplementation with diphenyl diselenide at 42, 28, and 14 days prior to calving and distributed in groups: diphenyl diselenide (DDG), dimethyl sulfoxide (DMSOG), and NaCl (NACLG) (average ± standard error, *n* = 6/treatment).

Parameters	Group	Values
Brix index (%)	NACLG ^1^	24.87 ± 1.21
	DMSOG ^2^	24.62 ± 0.80
	DDG ^3^	25.50 ± 1.54
IgG ^4^ (g/L)	NACLG ^1^	111.67 ± 11.18
	DMSOG ^2^	106.70 ± 12.83
	DDG ^3^	113.83 ± 12.29

Means followed by distinct capital letters refer to differences (*p* < 0.05) between groups (vertical; column). ^1^ NaCl group (NACLG); ^2^ Dimethyl sulfoxide group (DMSOG); ^3^ Diphenyl diselenide group (DDG); ^4^ Immunoglobulin G (IgG).

**Table 3 animals-14-00010-t003:** Composition of the mineral and vitamin supplement provided to Holstein calves from birth (day 0) to 70 days of life, distributed into different experimental groups.

Ingredient	Quantities/kg *
Vitamin A (IU/kg)	74,806
Vitamin D3 (IU/kg)	21,210
Vitamin E (IU/kg)	1090
Calcium (g/kg)	175.2
Chlorine (g/kg)	24.6
Crude protein (g/kg)	119.5
Ethereal extract (g/kg)	54.2
Lactose (g/kg)	70.5
Sodium (g/kg)	16.5
Biotin (mg/kg)	25
Chromium (mg/kg)	8.3
Cobalt (mg/kg)	4.4
Cooper (mg/kg)	354.8
Fluorine (mg/kg)	295
Iodine (mg/kg)	15
Magnesium (mg/kg)	1500
Manganese (mg/kg)	1068.2
Monensin (mg/kg)	700
Phosphorus (mg/kg)	8000
Potassium (mg/kg)	2500
Selenium (mg/kg)	6.5
Sulfur (mg/kg)	300
Zinc (mg/kg)	1022.1
*Saccharomyces cerevisiae* (CFU/kg)	1.0 × 10^11^

* minimal vitamin and mineral levels per kg of product.

## Data Availability

The data presented in this study are available on request from the corresponding authors (C.M.R. and M.L.R.L.).

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
