# Peer review of "Effect of Prepartum Maternal Supplementation with Diphenyl Diselenide on Biochemical, Immunological, and Oxidative Parameters of the Offspring"

_animals, 2023, doi:10.3390/ani14010010_

Round 1

Reviewer 1 Report

Comments and Suggestions for Authors

The results of the study are of interest to an international scientific audience (not only veterinarians, but also physiologists, doctors, and biochemists). The authors conducted original research, a labor-intensive long-term experiment. The data is reliable, correctly statistically processed, no obvious errors were noticed. However, the authors did not comply with international standards for the description (self-sufficiency, understandability of figures and tables without reading the material and methods) of statistical processing in figures and under tables. Research ethics complies with international standards, and the authors described it in some detail. The practical significance of the research is great.

It is unusual for me to read an article that combines results and discussion. However, journal standards probably do not prohibit this. Although the article would greatly benefit from having the results written separately and also divided into several subsections. It is advisable to write similar names of subsections in the discussion of this article. This would make the presentation of the material more rigorous and understandable.

I believe that once the specific deficiencies described below are addressed, this article can be recommended for publication.

Disadvantages of the manuscript.

1. In the second column of Table 2, you need to fully decipher the names of the groups and write their abbreviations in brackets. In the title of Table 2 you need to write in parentheses (average value + - standard error, repetition of the experiment = ...).

2. In the first column of Table 3 you need to write the characteristic comma unit of measurement. In the second column of this table you need to write only a number (without a unit of measurement).

3. In the note under table 4, you need to decipher the names of animal groups (from column 2).

4. The numbers in the first 6 lines of Table 4 must be rounded to the nearest tenth (both the mean and the standard error). I recommend rounding the last 3 rows of this table to thousandths (both the mean and the standard error). In the table header, “Days of Life” should not be above “Parameters” and “Group”. In the title of Table 4 you need to write in parentheses (average value + - standard error, repetition of the experiment = ...).

5. The same remarks apply to Table 5 and others. The abbreviations of the parameters (the first column of the table) must be deciphered under the table. Of the 6 parameters discussed in Table 5, the authors correctly rounded the first 3 parameters to hundredths, the fourth parameter should be rounded to thousandths, the fifth parameter should be rounded to whole numbers, and the sixth parameter should be rounded to tenths.

6. All fonts in Figure 1 must correspond in height to the size of the letters in the text of the article. The inscription above Figure 1 must be removed. On the ordinate and abscissa axis you need to write “Name of characteristic with a capital letter comma unit of measurement.” The comparison of samples in this figure is made incorrectly (at least, it is described incorrectly and does not correspond to the lines of Article 200-201). The reader will be interested to see whether there are differences between the values within the same graph over time and between different graphs in the same period of time. What is the repeatability of the experiment? You need to indicate this in the title of the drawing.

7. If the note under table 6 and under table 4 are the same, it is enough to write “see. table 4".

8. The first 3 rows of table 6 need to be rounded to tenths, the last 3 rows need to be rounded to whole numbers. The remaining errors are as in the previous tables.

9. There is no need to capitalize all the words in the title of the article (lines 416, 450, 453 and others).

10. Abbreviations of journal names are incorrect (lines 454, 462 and others).

Author Response

Caro revisor, 

Por favor, verifique o anexo.

Reviewer 2 Report

Comments and Suggestions for Authors

Line 23: such as weaning

23: Oxidative stress is a broad concept. Please, specify in which tissue the oxidative stress scenario takes place.

25: could be a valuable, positive or promising option.

36-37: please, mention how were dams supplemented. With (PhSe)2? Which kind? For how long?

41: calves on the DDG group

47: What is the conclusion of the article shown in the abstract? What is the take-home message?

60-64: Please, mention if the fetus cannot receive any kind of oxidative stress agent during gestation, such as ONOO-, (-OH), O2-, etc. It will be interesting to read that argurment in order to understand the importance of ROS after birth, in the case the calf was not exposed to radical agents during gestation.

64-68: This sentence is not fully clear, please re-write it.

80-84: Is this argument also true for young calves? It is to say, when calves did not developed their rumen?

100: depressive behavior does not seem to be a relevant benefit to the article.

102: when starting a new paragraph, do not begin the sentence with “it”. Please, write Se instead of “it”

112: aimed at assessing

114: Holstein calves

117: Did authors have the approval from an ethical committee to develop this type of study? Please, mention the authorization of some ethical administration in Brazil, if present. Also, the section 2.2 should be the first section in materials and methods.

127: n= 18

130: please, change “mothers” by “dams”

142: Are the ingredients shown in as-fed or dry-matter basis? How was the selenium content in the control and treatment diet? How was the quality of the diet in terms of CP, Energy (MEm, MEl), mineral content (specially Selenium), fat, etc?

153: Location of Vee Gee headquarters? Please, review the other companies mentioned throughout the manuscript (i.e., BD vacutainer)

157: “p” not “p”. Are the values ± standard errors or standard deviation?

159-161: What was the temperature of the milk provided to calves?

168: The experimental design should be in the same section than “animals”. Also, the “animals” section presents information related to the diet. I strongly suggest to create one section including experimental desing that includes animals and diet.

169-172: How many animals per treatment? Why DDG group received DMSO? Should not have to receive DDG?

175-178: Treatment distribution of calves is not clearly expressed in the manuscript. Please, re-write

181: remove “and again”.

192: please, do not begin a sentence with an abbreviation

234: I suggest that, if there is no difference in a parameter analyzed, it is better to not include a letter to differentiate means (i.e., Weight at day 0, author can remove the Aa since all the treatments are the same and it will be easier for the reader)

234: Body weight instead of weight

255: (14, 28, 42, 56, and 70; p < 0.05).

270-274: This argument should not be included since we, as scientists, rely on statistical analysis to understand difference in the natural world.

295-297: Please, add reference

301: How can authors show treatment x time interactions in tables? I suggest to convert the tables into graphs, and that will help authors to show the interactions.

313: among all groups

413: Not all the citations have the doi number

Comments on the Quality of English Language

Please, review the comments.

Author Response

Dear reviewer,

Reviewer 3 Report

Comments and Suggestions for Authors

This study shows the calf-rearing phase in dairy production is critical due to their immature immune systems and susceptibility to disease. Stressors like weaning can lead to oxidative stress as their antioxidant defenses are also underdeveloped. To combat this, researchers are exploring prepartum maternal selenium supplementation, using diphenyl diselenide for its anti-inflammatory and antioxidant effects, and its safe subcutaneous administration. This method has shown to enhance passive immunity transfer and humoral immunity in calves, indicating the compound’s potential immunomodulatory benefits. However, there are some issues before publication.

1. Line 761-766: Clarify the comparison between the direct administration of (PhSe)2 to calves and maternal supplementation during prepartum. Specify if the effects measured are directly comparable or if there are different metrics or conditions considered.

2. Line 779-785: The section discussing the measurements for evaluating passive immunity transfer in calves could benefit from additional context or preliminary data that demonstrates the baseline levels for these immunoglobulins and proteins in healthy calves for comparison.

3. Line 819-822: The significant changes in IgG levels in the DD group need more detailed explanation. Discuss potential mechanisms or hypotheses on why these changes occur and their implications for calf health and development.

4. Line 823-830: When discussing Se bioavailability and its impact on the calves, consider incorporating a discussion on the possible variations due to breed differences, feeding practices, or environmental factors that could influence the results.

5. Line 889-890: Expand on the role of selenium in the pinocytosis process and its influence on immunoglobulin absorption. Provide more detailed evidence or references to support the claim.

6. Line 1189-1192: Address the lack of established reference values for TBARS and GSH in calves by discussing how this affects the interpretation of the results and the study's conclusions. Suggest potential benchmarks or future research necessary to establish these reference values.

Author Response

Dear Reviewer, 

Round 2

Reviewer 2 Report

Comments and Suggestions for Authors

I suggest to transform the tables into graphs, in order to see the results more clearly.

61: milk production?

131: be consistent in the terminology. Use “eighteen” or “18”.

143: I suggest writing the unit next to the chemical composition (i.e., CP, %DM; ME, Mcal/kg) and not doing another column in the table, such as in Table 4.

Author Response

Dear Reviewer, 

Response to Reviewer 2 Comments

1. Summary

Dear reviewer, thank you very much for taking the time to review this manuscript. Please find the detailed responses to comments below and the corresponding revisions/corrections highlighted in red in the re-submitted manuscript file.

2. Questions for General Evaluation

Reviewer’s Evaluation

Response and Revisions

Does the introduction provide sufficient background and include all relevant references?

Yes

Responses to reviewer comments are exposed point-by- point in the document below.

Are all the cited references relevant to the research?

Yes

Is the research design appropriate?

Yes

Are the methods adequately described?

Yes

Are the results clearly presented?

Can be improved

Are the conclusions supported by the results?

Yes

3. Point-by-point response to Comments and Suggestions for Authors

I suggest to transform the tables into graphs, in order to see the results more clearly.

Comments 1: 61: milk production?

Response 1: Thank you for pointing this out. We agree with this comment. Therefore, we include the information and highlighted it in red.

Comments 2: 131: be consistent in the terminology. Use “eighteen” or “18”.

Response 2: Dear reviewer, thank you for pointing this out. We standardized the sentences with "eighteen", the terminology "18" only continued to be used in two instances, where it appears as "n=18" (lines 127 and 153).

Comments 3: 143: I suggest writing the unit next to the chemical composition (i.e., CP, %DM; ME, Mcal/kg) and not doing another column in the table, such as in Table 4.

Response 3: Thanks for pointing this out. We agree with this comment and made the change in tables 1 and 3, deleting the second column (unit), keeping only two columns, and indicating the unit of ingredient/chemical composition next to the items listed.

4. Response to Comments on the Quality of English Language

Point 1: The reviewer marked the option (x) English language fine. No issues detected

Response 1: Dear reviewer, in the absence of new comments related to the quality of the English language, we take this opportunity to thank you again for your previous comments, as your collaboration was essential to improving this work.

5. Additional clarifications

Dear reviewer, we appreciate your comment suggesting the replacement of tables with graphs, and we agree. Therefore, we removed tables 4, 5, 6 and 7 from the revised manuscript and included figures 1, 2 and 3, which have the results of the tables grouped together. They describe the comparisons between treatment, moment, and treatment x moment interaction. We would like to thank the reviewer again for his insightful comments, and the editor for considering our work for review.

Reviewer 3 Report

Comments and Suggestions for Authors

Since the author has addressed all the concerns, the paper is ready to be published.

Author Response

Dear Reviewer,

Response to Reviewer 3 Comments

1. Summary

Dear reviewer, thank you very much for taking the time to review this manuscript and nominate it for publication. As there was no need to respond to comments in this round, we chose to present our responses to the review in item 2 (questions for general evaluation, response and revisions). However, some revisions/corrections corresponding to the comments of other reviewers are highlighted in red in the resubmitted manuscript file.

2. Questions for General Evaluation

Reviewer’s Evaluation

Response and Revisions

Does the introduction provide sufficient background and include all relevant references?

Yes

Yes

Are all the cited references relevant to the research?

Yes

Yes

Is the research design appropriate?

Yes

Yes

Are the methods adequately described?

Yes

Yes

Are the results clearly presented?

Yes

Yes

Are the conclusions supported by the results?

Yes

Yes

3. Point-by-point response to Comments and Suggestions for Authors

Since the author has addressed all the concerns, the paper is ready to be published.

4. Response to Comments on the Quality of English Language

Point 1: The reviewer marked the option (x) English language fine. No issues detected

Response 1: Dear reviewer, in the absence of comments related to the quality of the English language, we take this opportunity to thank you again for your previous comments, as your collaboration was essential to improving this work.

5. Additional clarifications

As the reviewer left only one comment stating that their concerns were addressed and recommending the work for publication, we not have any additional information to mention. We would like to thank the reviewer again for his insightful comments and considerations, and the editor for considering our work for review.